



# Inter-comparison of Atmospheric Trace Gas Dispersion Models: Barnett Shale Case Study

Anna Karion[1], Thomas Lauvaux[2], Israel Lopez Coto[3], Colm Sweeney[4], Kimberly Mueller[1], Sharon Gourdji[1], Wayne Angevine[4,5], Zachary Barkley[2], Aijun Deng[6], Arlyn Andrews[4], Ariel Stein[7], and James Whetstone[1]

[1]Special Programs Office, National Institute of Standards and Technology, Gaithersburg, Maryland, USA

[2]Department of Meteorology, The Pennsylvania State University, University Park, Pennsylvania, USA

[3]Fire Research Division, National Institute of Standards and Technology, Gaithersburg, Maryland, USA

[4]Earth System Research Laboratory, National Oceanic and Atmospheric Administration, Boulder, Colorado, USA

[5]Cooperative Institute for Research in Environmental Sciences, University of Colorado, Boulder, Colorado, USA

[6]Utopus Insights, Valhalla, New York, USA

[7]Air Resources Laboratory, National Oceanic and Atmospheric Administration, College Park, Maryland, USA

*Correspondence to:* Anna Karion (Anna.Karion@nist.gov)

**Abstract.** Greenhouse gas emissions mitigation requires understanding dominant processes controlling fluxes of these trace gases at increasingly finer spatial and temporal scales. Trace gas fluxes can be estimated using a variety of approaches that translate
observed atmospheric species mole fractions into fluxes or emission rates, often identifying the spatial and temporal characteristics of the emissions sources as well. Meteorological models are commonly combined with tracer dispersion models to estimate fluxes using an inverse approach that optimizes emissions to best fit the trace gas mole fraction observations. One way to evaluate the accuracy of atmospheric flux estimation methods is to compare results from independent methods, including approaches in which different meteorological and tracer dispersion models are used. In this work, we use a rich data set of atmospheric methane
observations collected during an intensive airborne campaign to compare different methane emissions estimates from the Barnett Shale oil and natural gas production basin in Texas, U.S.A. We estimate emissions based on a variety of different meteorological and dispersion models. Previous estimates of methane emissions from this region relied on a simple model (a mass balance analysis) as well as on ground-based measurements and statistical data analysis (an inventory). We find that in addition to meteorological model choice, the choice of tracer dispersion model also has a significant impact on the predicted downwind
methane concentrations given the same emissions field. The dispersion models tested often under-predicted the observed methane enhancements with significant variability between different models and between different days. We examine possible causes for



this result and find that the models differ in their simulation of vertical dispersion, indicating that additional work is needed to evaluate and improve vertical mixing in the tracer dispersion models commonly used in regional trace gas flux inversions.

## 1 Introduction

As climate research advances and climate models attempt to predict the effect of increasing atmospheric abundance of greenhouse
gases (GHG) on the global climate with smaller uncertainties, reducing uncertainty in our understanding of the carbon cycle becomes increasingly relevant, e.g., for carbon dioxide ($CO_2$) (Le Quéré et al., 2017) and methane ($CH_4$) (Saunois et al., 2016). GHG emissions mitigation also requires understanding dominant processes affecting emissions of these gases, often at smaller regional or even urban scales. Approaches using atmospheric observations to estimate surface emissions of $CH_4$, specifically, have been implemented at global (e.g. Bruhwiler et al., 2017) and regional (e.g. Jeong et al., 2016) scales to evaluate reported emissions
and inventories. Many of these studies have shown a gap between inventory and atmospheric-based estimates (e.g. Miller et al., 2013; Brandt et al., 2014). But investigations of $CH_4$ and other trace gas fluxes that rely on atmospheric observations also depend on accurate simulation of atmospheric transport to estimate emissions. In global studies, there have been efforts to compare different transport and inverse models in order to understand uncertainties and differences in their derived flux estimates (e.g. Gurney et al., 2004). But these efforts remain limited, especially at the regional (e.g. Schuh et al., 2013) and local scales.

Flux estimation methods have been applied at regional and local spatial scales, including in urban areas and oil and gas production fields, using dense airborne or ground-based observation networks or intensive campaign-based observations (Staufer et al., 2016; Lauvaux et al., 2016; Cui et al., 2015). At these scales, additional errors can arise from since the meteorological models need to faithfully replicate conditions at finer spatial and temporal scales (Deng et al., 2017; Lac et al., 2008). Meteorological models
yield averaged transport and turbulence characteristics (e.g., mean wind velocities and turbulent kinetic energy) that are then used to transport and disperse a tracer, often in a separate off-line model. The tracer dispersion model simulates concentrations at the observation location, either by transporting emissions forward in time to the observation point or performing a backwards-in-time calculation of the influence region ("footprint") of an observation and convolving the influence function with an emissions field. These two methods are nominally equivalent, and both rely on underlying meteorological fields to drive the calculation. Mesoscale
meteorological models, such as the Weather Research and Forecast model (WRF), are often coupled with Lagrangian tracer dispersion models, (e.g., FLEXible PARTicle Dispersion Model (FLEXPART), Stochastic Time-Inverted Lagrangian Transport (STILT), Hybrid Single Particle Lagrangian Integrated Trajectory (HYSPLIT), Lagrangian Particle Dispersion Model (LPDM), Numerical Atmospheric-dispersion Modelling Environment (NAME), and others) for use in inversion methods at scales of tens to hundreds of kilometers and hours to weeks (Cui et al., 2015; Lauvaux et al., 2016). These Lagrangian tracer dispersion models
are typically run off-line, i.e., using archived meteorological model output and parametrizations of vertical and horizontal dispersion to simulate tracer transport.. The Eulerian WRF model itself also has the capability to include chemical or passive tracer



dispersion (WRF-Chem) that is run in-line with the meteorological model, transporting trace gas emissions forward in time. WRF-Chem has been shown to simulate $CH_4$ observations successfully in previous work (Ahmadov et al., 2015; Barkley et al., 2017).

Some investigation of meteorological model differences has resulted in the conclusion that differences in parametrizations in the mesoscale meteorology (i.e., in WRF or other model), including differences in initial and boundary conditions (Angevine et al., 2014), are more significant than the choice of tracer dispersion model. For example, Hegarty et al. (2013) compared the performance of STILT, HYSPLIT, and FLEXPART against historical tracer experiment data and found generally good agreement between these Lagrangian dispersion models when driven by the same WRF fields. However, a comparison of online Eulerian and off-line Lagrangian (WRF-STILT) simulations where the Lagrangian simulations were driven by the archived meteorology from the Eulerian simulation suggested differences in vertical dispersion between these models during some time periods tested (Pillai et al., 2012). Lauvaux et al. (2012) also found that there were some cases (a few days per month in that study) where the Eulerian (WRF-Chem) and the Lagrangian (LPDM) modeled concentrations showed large differences (when run using the same emissions field) and that these cases were associated with stable conditions in the lower atmosphere. These regional studies suggest that the tracer dispersion characteristics (and not only the underlying meteorological fields) can and do affect simulated concentrations.

In March 2013 and October 2013, an aircraft-based campaign was conducted with the goal of quantifying $CH_4$ emissions from oil and gas operations in the Barnett Shale region of Texas, U.S.A. The airborne data set was used by Karion et al. (2015) and Smith et al. (2015) to estimate total emissions of $CH_4$ and ethane in the region using a mass balance estimation (MBE) method. The large domain, spatial wind variability, and long air mass transport times (5 h – 12 h) in the Barnett Shale study necessitated the use of model-based averaged winds in the MBE from WRF. The campaign also included numerous ground-based measurements (Harriss et al., 2015; Marrero et al., 2016), meteorological modeling (Lauvaux et al., 2013), and the construction of a detailed inventory of $CH_4$ sources in the 25-county Barnett Shale area (Lyon et al., 2015). A second statistical analysis was performed to adjust the inventory based on ground-based and other local measurements. The new inventory was shown to agree well with the top-down aircraft-based MBE result (Zavala-Araiza et al., 2015).

This data set presents a unique opportunity to test the ability of a meteorological model coupled with tracer dispersion to replicate observed concentrations given a realistic source distribution of emissions. The inventory developed by Zavala-Araiza et al. (2015) also includes an uncertainty estimate, in the form of 95% confidence intervals that are spatially resolved. In addition to the availability of this high-quality inventory for the study region and time frame, meteorological conditions during the Barnett airborne campaign were considered generally favorable for transport models, with relatively flat terrain, clear weather, and no sharp changes in wind direction or weather conditions during the 12 hours prior to each flight, making the Barnett campaign flights ideal candidates for evaluating both transport and tracer dispersion models. In this study, we use these flight observations to investigate the impact of meteorological and tracer dispersion model differences on estimated fluxes in a regional domain.



To this end, a series of transport and dispersion model runs were conducted to simulate CH₄ enhancements. We include simulations using (1) identical meteorological model output but different tracer dispersion models as well as (2) simulations using both different meteorological and dispersion models. To further examine the causes of differences between various model runs on two specific

flight days, we then compare forward-in-time simulations using one Lagrangian tracer dispersion model (HYSPLIT) and the Eulerian (WRF-Chem) model, using identical fluxes and meteorology to better diagnose the cause of the differences in tracer dispersion. Lastly, we use the original set of model runs to estimate fluxes using a simple inventory scaling as well as a classical Bayesian inversion.

Section 2 describes the methodology used, including the various transport and dispersion model combinations and flux estimation methods investigated for the study. Sections 3.1–3.2 focus on the comparison of simulated CH₄ enhancements between different models and observations, including comparison of forward-run models on two flight days. Sections 3.3 and 3.4 examine the impact of meteorological/dispersion model choice on emissions estimates. In Sect. 4 we discuss causes and impacts of the differences between dispersion models, with overall conclusions in Sect. 5.

## 2 Methods

### 2.1 Experimental domain and observations

Figure 1 shows the region map with the 25-county outline of the Barnett shale as defined by the Texas Railroad Commission, along with the 0.1 degree gridded Zavala-Araiza (2015) inventory, referred to here as the Z-A inventory. The aircraft typically sampled in the planetary boundary layer (PBL), with between one and three vertical profiles per flight for PBL depth determination, and

conducted between one and five transects downwind of the area with most dense natural gas production. We refer to individual flight days by the 8-digit date, YYYYMMDD, and CH₄ observations and model output are reported here in nanomoles per mole of dry air (ppb). Figure 1 shows flight tracks from two of the flight days, 20131019 and 20131028, chosen as examples here and later in the analysis. The mean horizontal wind for 20131028 was from the south, so that the downwind transects are visible in the north of the domain; the opposite wind direction on 20131019 meant that downwind transects were in the south (Fig. 1). Downwind

transects did not always cover the entire downwind extent, because observed CH₄ returned to background levels before that was reached, indicating that emissions from the western sector were not significant (more discussion can be found in Karion et al. (2015)). These downwind transects were used for the MBE in Karion et al. (2015) and are used also in much of the analysis presented here. We focus on the eight flights used in Karion et al. (2015), for ease of comparison, but also because these are the flights that best covered the region and sampled downwind of the main emissions area.





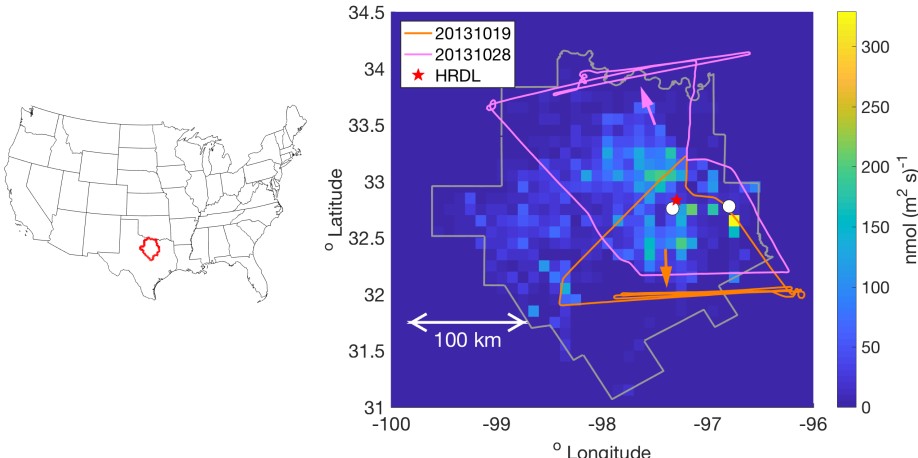

**Figure 1. Left:** Location of the 25-county Barnett Shale domain (red) within the US. **Right:** The 25-county domain covered by the Z-A inventory (gray outline) and colored by the magnitude of the inventory $CH_4$ fluxes (shown at the 0.1 degree resolution used here). Two sample flight tracks are indicated, from 20131019 and 20131028, with arrows of the same color indicating the average wind direction on each day. Cities of Fort Worth and Dallas are indicated by white circles. The red star near Fort Worth indicates the location of the High-Resolution Doppler Lidar (HRDL), whose measurements were used to evaluate the WRF model wind speed and PBL depth.

## 2.2 Meteorological and dispersion models – backward/footprint models

Two different mesoscale meteorological models were investigated for this work: the North American Mesoscale (NAM) 12 km model (archived fields available at ftp://arlftp.arlhq.noaa.gov/nams/), and multiple versions of the Weather Research and Forecast (WRF) model. WRF is commonly used in regional-scale trace gas flux estimation, often run for specific time periods and regions at relatively high spatial resolution (1 km). The 12 km NAM model is operationally available and has also been used for flux inversions (Hu et al., 2016), so here we evaluate its performance relative to the more computationally expensive customized implementations of the WRF model.

The primary meteorological model used in this analysis is WRF with four-dimensional data assimilation (FDDA) reanalysis, whose wind fields were used in the MBE in Karion et al. (2015). We refer to this implementation as WRF in Table 1 and elsewhere. Evaluation of WRF wind and PBL depth using High-Resolution Doppler Lidar (HRDL) and aircraft observations is in the Supplementary material, Sect. S1, Fig. S1–S3, and Tables S1 and S2, with configuration details in Sect. S2.1. For the Lagrangian dispersion models in this study we used only the innermost two domains, with 1 km and 3 km spatial resolution and 20 minute and 1 hour temporal resolution, respectively, as all the flight data and footprints were contained within their grids (see Karion et al. (2015) Supplementary Information for maps of WRF domains).



For all eight flights (three in March and five in October 2013), we ran three different Lagrangian tracer dispersion models in backward-time mode driven by these same WRF meteorological fields: NOAA/ARL's HYSPLIT model (Stein et al., 2015; Draxler and Hess, 1997), the Stochastic Time-Inverted Lagrangian Transport (STILT) model (Lin et al., 2003), which is based on the

HYSPLIT model, and the Lagrangian Particle Dispersion Model (LPDM) (Uliasz, 1994). The HYSPLIT model was also run using NAM wind fields. In backward mode, a group of particles representing mass-less air parcels is released from a receptor location and time corresponding to an individual $CH_4$ observation, and the particles are tracked backwards in time, driven by both mean winds and turbulence characteristics from input meteorological fields. All the Lagrangian models produce "footprints", or influence functions, for each receptor, with units $\left( \frac{ppm}{\mu mol(m^2 s)^{-1}} \right)$ (Fig. S4). The footprints are functions of the residence

time of each particle over a given spatial grid cell (Lin et al., 2003), and can be convolved (i.e., multiplied on a grid-cell by grid-cell basis and summed over the domain) with fluxes to simulate $CH_4$ enhancements over background at the receptor point (i.e., measurement location). For two flights, 20131019 and 20131028, additional dispersion model configurations were investigated using HYSPLIT and STILT by varying parameter choices within the models. Model details and results of these dispersion model parameter choices are discussed in Sect. S2.3 and Fig. S5; none of the parameters tested within STILT or HYSPLIT had a

substantial effect on the modeled $CH_4$ mole fraction enhancements.

Along with the above, we also investigated two additional transport and dispersion model combinations to provide more comparison points with other commonly used models for trace gas flux inversions. First, we compare with the WRF-STILT configuration used for NOAA's Carbon-Tracker Lagrange regional inverse modeling framework (CT-L,

https://www.esrl.noaa.gov/gmd/ccgg/carbontracker-lagrange/; Sect. S2.7). A second model comparison with another variation of WRF coupled with FLEXPART (designated WRF2-FP), was run for the October 2013 flights (5 flights; Sect. S2.6). Table 1 indicates the specific combinations used for this study, with more details on how each was configured in the Supplementary material, Sect. S2. All footprints were run every 30 seconds (~2.1 km) along the flight paths, gridded over the same domain at 0.1 x 0.1 degrees (unless noted), and have hourly resolution backward in time.




**Table 1. Various meteorological and dispersion model combinations performed for this study. Details of specific parameter choices are in the text and Sect. S2. Models in the first seven rows (gray shading) were used in the flux determination in Sect. 3.**

| Flights | Meteorological Model | Dispersion Model | Direction in time |
|---|---|---|---|
| All | WRF | HYSPLIT | Backward |
| All | WRF | STILT | Backward |
| All | WRF | LPDM | Backward |
| All | NAM | HYSPLIT | Backward |
| All except 20130325 | WRF-CT-L | STILT | Backward |
| 201310XX (5 flights) | WRF2 | FLEXPART (FP) | Backward |
| 20131019, 20, 25, 28 | WRF-Chem | WRF-Chem (Eulerian) | Forward |
| 20131019, 20, 25, 28 | WRF-Chem | HYSPLIT | Forward |
| 20131019, 20131028 | WRF | HYSPLIT with parameter changes | Backward |
| 20131019, 20131028 | WRF | STILT with parameter changes | Backward |
| 20131028 | WRF | HYSPLIT with 0.04 deg resolution | Backward |
| 20131028 | WRF | HYSPLIT with 5000 particles | Backward |

**2.3 Convolutions of footprints with the inventory**

The Zavala-Araiza (Z-A) CH₄ inventory was provided to us at 0.1 degree resolution. Briefly, it was constructed using a detailed accounting of activity in the region for the month of October 2013 with updated emission factors based on ground-based measurements and statistical analysis of the long-tailed distribution typically associated with oil and gas production basin emissions (Lyon et al., 2015; Zavala-Araiza et al., 2015; Zavala-Araiza et al., 2017). The detailed construction of this inventory for the time

period covered by the October flights and its consistency with the MBE result gives high confidence in its accuracy. This inventory was also used as a benchmark for estimating error in the EPA gridded 2012 inventory (Maasakkers et al., 2016). The inventory only contains non-zero emissions within the 25-county Barnett region (Fig. 1), with sources outside the 25 counties not included. We convolve the Z-A inventory with footprints from the backward Lagrangian model runs for each flight, assuming emissions are constant in time, so that the sum of the footprint strength over all hours in each 0.1x0.1 degree cell is multiplied by the inventory

value at that cell, and all the cells are also summed for each receptor, or observation point. The resulting modeled CH₄ mole fraction enhancement is calculated in this manner for each flight track and is compared with the observed enhancement (averaged at 30 s intervals to match the model). We also compared enhancements using the EPA gridded inventory for 2012 (Maasakkers et al., 2016), with no significant difference in results, so they are not reported here. In this analysis, all CH₄ mole fractions are expressed in nmol of CH₄ per mole of dry air (ppb). For the comparison and all the subsequent analysis, the background mole

fraction that was used in the MBE, derived from the edges of the plume observed downwind of the emissions region (Karion et al., 2015), has been subtracted from all the observations in each flight. This was done to maximize the comparability between the





MBE and modeling results; any errors in background determination, including the possibility of upwind sources affecting enhancements, would exist in both analyses.

### 2.4 Forward simulations

In addition to the backward-in-time, Lagrangian footprint-based simulations of CH4 enhancements described above, two different
forward models were investigated, one online Eulerian (WRF-Chem) and one off-line Lagrangian (WRF-HYSPLIT). This step is critical in identifying that the differences are not affected by whether the dispersion model is run forward or backward in time, but rather hinge on the parametrization of vertical mixing. These forward HYSPLIT runs were driven by the WRF-Chem 3 km hourly meteorological fields while the backward/footprint runs were driven by the WRF-FDDA 1 km (20 minute) and 3 km (hourly) nested fields. In both HYSPLIT and WRF-Chem, the CH4 emissions from the Z-A inventory were tracked forward in time (with
no chemistry), with concentrations simulated as full 4-dimensional fields. We extract the time series along the flight path by sampling the 4-dimensional mole fraction fields at 30 s intervals at the time and (interpolated) location of the observations. Further details on the configuration of these forward runs are in Sect. S2.2 and Sect. S2.4; comparisons of forward HYSPLIT runs with the equivalent backward runs are shown in Fig. S6.

### 2.5 Flux estimation

We use two methods for estimating CH4 fluxes in the 25-county Barnett region, using the eight flight days' observations coupled with the model-simulated CH4 enhancements. We employ both an inventory scaling method (McKain et al., 2015; Kort et al., 2008) and a classical Bayesian flux inversion, as described below. These are two methods commonly employed for trace gas flux estimation. We note that for the flight on 20131019, the WRF model under-predicted the PBL depth (evaluated from the aircraft observations), so that one of the downwind segments that was flown at relatively higher altitude (but still within the PBL), shows
no little to no enhancement in any of the models driven by WRF, because it is above the model PBL. These points and this entire segment were omitted from all the analyses.

#### 2.5.1 Inventory scaling method

For the inventory scaling estimation method, we average the observed CH4 enhancements along the downwind transects identified for the MBE in Karion et al. (2015). A single background value is subtracted from the observations, determined from Karion et
al., with an average value used in the cases where the background varied across the plume. Thus, we include negative enhancements (caused by either transport error or background error, not CH4 sinks) to get an appropriate average enhancement for each transect ($\Delta CH_{4,OBS}$). A corresponding average enhancement from the model simulations is calculated for each downwind segment ($\Delta CH_{4,MOD}$) (i.e., from the either the convolutions of the footprints with the inventory or the forward runs, depending on the model). The ratio of observed to model average segment enhancements is the scaling factor on the Z-A inventory required to





match the average enhancement in the observations (or the new emissions estimate ($E_{IS}$) divided by the Z-A inventory ($E_{Z-A}$) as shown in Eq. (1) below).

$$F_{scaling} = \frac{E_{IS}}{E_{Z-A}} = \frac{\Delta CH_{4,OBS}}{\Delta CH_{4,MOD}}$$

(1)

We estimate an uncertainty on the scaling factor using the relative uncertainty in $\Delta CH_{4,OBS}$, which is in turn based on the background uncertainty (from Karion et al.). There is no uncertainty included from the transport model. We assume a single scaling factor for each flight, i.e., assuming that the scaling factor applies across the entire domain, even in cases when the flight did not cover the entire region. This inventory scaling method is applied using seven transport/dispersion configurations (the first seven rows of

Table 1) for all the flights for which they were available.

### 2.5.2    Bayesian inversion

A classical Bayesian inversion approach was taken to estimate emissions using each individual flight separately. The following equation (Tarantola, 2004) was used to solve for both the posterior fluxes in the 25-county domain ($\hat{\mathbf{x}}$) and their corresponding posterior uncertainties, $\mathbf{A}$:

$$\hat{\mathbf{x}} = \mathbf{x}_b + \mathbf{B}\mathbf{H}^T(\mathbf{H}\mathbf{B}\mathbf{H}^T + \mathbf{R})^{-1}(\mathbf{y} - \mathbf{H}\mathbf{x}_b),$$    (2)

$$\mathbf{A} = \mathbf{B} - \mathbf{B}\mathbf{H}^T(\mathbf{H}\mathbf{B}\mathbf{H}^T + \mathbf{R})^{-1}\mathbf{H}\mathbf{B}.$$    (3)

In the formulation above, $\mathbf{B}$ represents the prior error covariance matrix, $\mathbf{H}$ is the sensitivity matrix, i.e., the matrix of footprints, $\mathbf{R}$ is the model-data mismatch covariance matrix, and $\mathbf{z}$ represents the vector of observations after the background mole fraction has been subtracted (Tarantola, 2004; Lauvaux et al., 2012). Unlike the scaling analysis, here we use all the flight observations, not only the downwind transect portions. $\mathbf{H}$ is obtained from the footprints of the WRF-HYSPLIT model runs for each flight (here we only perform inversions using this one model to illustrate how the inversion result compares to the inventory scaling); we obtain

$\mathbf{R}$ and $\mathbf{B}$ separately for each flight (described further below and in Sect. S3). The Z-A inventory was used as the prior ($\mathbf{x}_b$) for all the Bayesian inversions. Equations (2) and (3) are solved for each flight separately, and we sum the posterior fluxes over the area of the inventory (the 25-county domain) and convert to a mass flux in metric tons per hour (t h$^{-1}$), which can be directly compared to the total inventory emissions.

After an initial investigation of the impact of choice of the magnitude of the variances along the diagonal of the $\mathbf{R}$ and $\mathbf{B}$ matrices (Sect. S3 and Fig. S7), we employ a $\mathbf{B}$ matrix that is ten times the reported uncertainty of the inventory at each domain grid cell



(Fig. S8, lower right), with no off-diagonal terms. Two additional **R** matrices were also investigated, one estimated using a Restricted Maximum Likelihood (RML) method (Michalak et al., 2005), and one using the variance of enhancements predicted by the various transport and dispersion models. Details on the construction of these matrices are found in Sect. S3, with discussion of the consequences of these choices for **R** and **B** is in Sect. 3.4.

**3  Results**

**3.1  Modeled CH$_4$ enhancements using the Z-A inventory**

We choose to first focus on two of the eight flights in detail because they shed light on specific model sensitivities. In the second part of our analysis, we put these results into perspective by comparing these two flights to the other six flight days. Concerning the choice of the two flights, the first one (20131019) was selected because it contained the most downwind transects (five) at four

different altitudes and had the most intensive sampling of the downwind plume. The second (20131028) was chosen because the initial model configuration yielded the largest discrepancy with the observed enhancements. This discrepancy indicates that there is a problem specific to the transport and dispersion model as the errors in assumed emissions could not account for an underestimate of enhancements to this degree. We compare simulated enhancements along the flight paths from the various meteorological and tracer dispersion model combinations to observations (after background is subtracted) in Fig. 2 for the flight

on 20131019 and Fig. 3 for 20131028. The correspondence in time of the modeled enhancements with the flight observations in the time series on both days is quite good, with the CH$_4$ plume being represented in the correct location and with the correct overall structure in most models. The coefficient of determination ($R^2$) for each model for each flight is reported in Table S3.

In WRF-Chem, the emissions are trapped closer to the ground than in the other models, including those runs using archived wind

fields from the same WRF simulation. On 20131019, the aircraft flew close to the top of the WRF PBL, and in WRF-Chem, enhancements were low when sampled at the flight altitude (Fig. 2, dark pink). However, when the model was sampled 200 m lower in altitude, the enhancements matched observations significantly better (Fig. 2, light pink). On 20131028, we also found that WRF-Chem exhibited a vertical gradient in CH$_4$ such that enhancements were larger when sampled closer to the ground (again 200 m lower than the aircraft flight altitude), matching the observations better (Fig. 3, light pink). WRF-HYSPLIT did not show

any significant difference in enhancement when sampled lower on either day, because the emissions were well-mixed in the PBL in that model, and the flight was conducted inside the mixed layer, except for the very top transect at ~890 m above ground altitude. Vertical mixing in the models for these two flights is investigated further in Sect. 3.2.

As expected, the differences between dispersion models run with different meteorology (top panels in Figs. 2 and 3) are large.

However, differences between dispersion models run using identical meteorology (bottom panels) are also significant, particularly on 20131028. On this day, WRF-Chem (at flight altitude) and WRF-LPDM showed larger enhancements than the other models,



coming closer to matching observations, but still generally smaller than observed. When sampled at the lower altitude within its

PBL, WRF-Chem showed significantly larger enhancements, closer to observations.

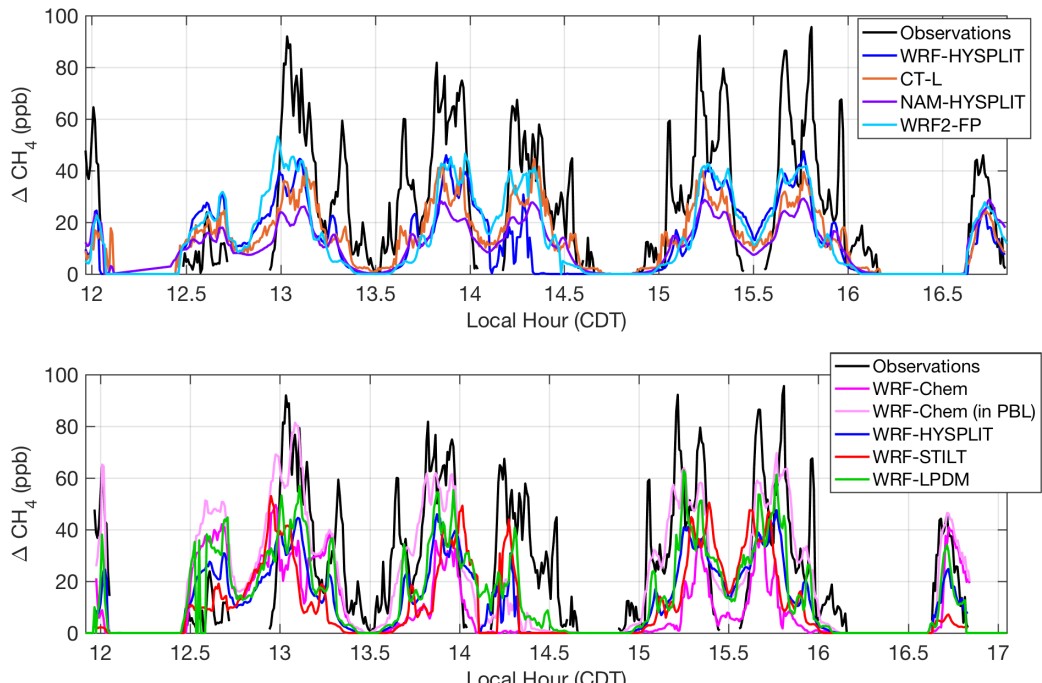

5    **Figure 2. Modeled (colored lines) and observed (black lines) CH$_4$ enhancements along flight track on 20131019: (top) models using different transport and dispersion, and (bottom) models using identical meteorological fields (WRF) and different dispersion. Light pink line indicates the values when WRF-Chem is sampled 200 m below the aircraft altitude. WRF-HYSPLIT (dark blue) is included in both plots as a reference point. Local time is Central Daylight Time (CDT).**





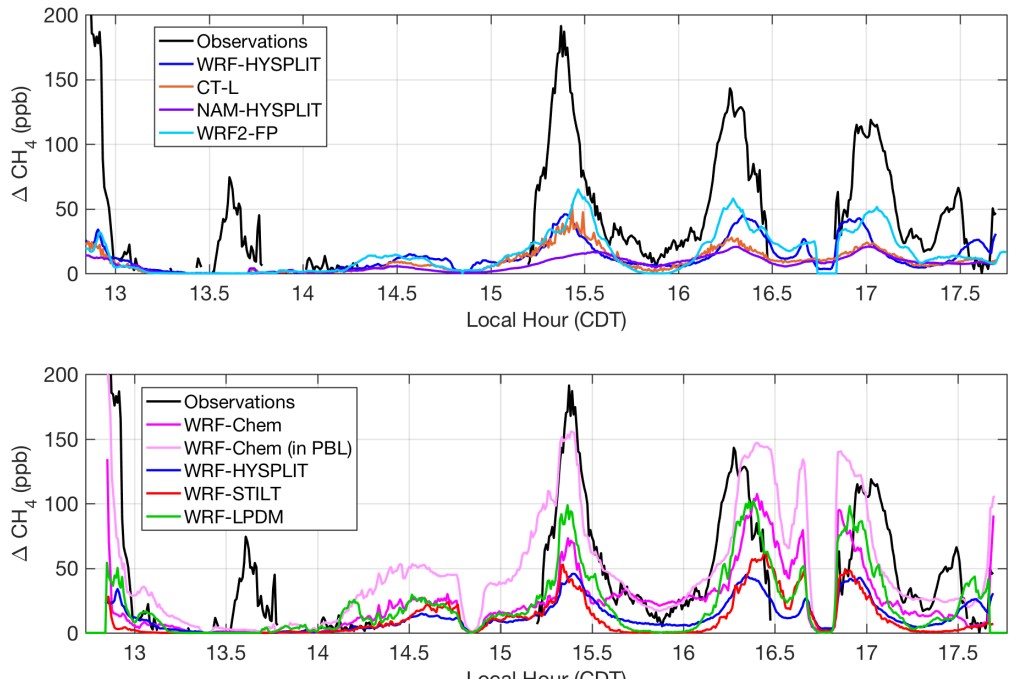

**Figure 3.** Modeled (colored lines) and observed (black lines) CH₄ enhancements along flight track on 20131028: (top) models using different transport and dispersion, and (bottom) models using identical meteorological fields (WRF) and different dispersion. Light pink line indicates the values when WRF-Chem is sampled 200 m below the aircraft altitude. WRF-HYSPLIT (dark blue) is included in both plots for a reference point. Local time is Central Daylight Time.





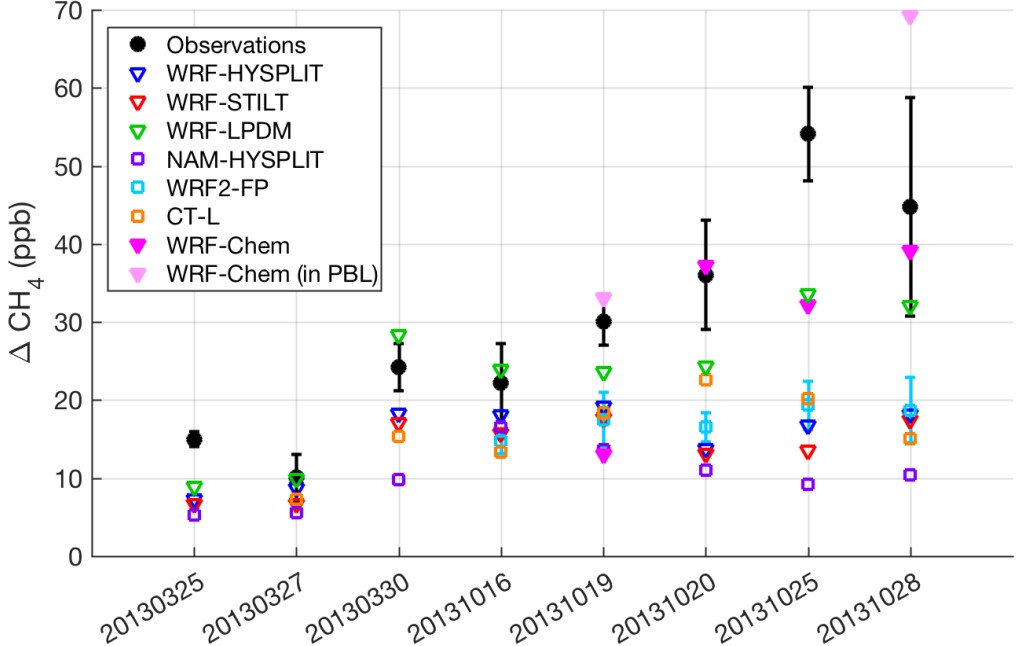

**Figure 4. Average CH$_4$ enhancement in downwind transects from various meteorology/dispersion models (open squares), different dispersion models using identical WRF meteorology (open triangles), WRF-Chem forward runs (magenta triangles at flight altitude, light pink 200 m below flight altitude for 20131019 and 20131028), and observations (black circles). Error bars on the WRF2-FP (FLEXPART) runs indicate the standard deviation of results from four different WRF2 configurations (three on 20131016), as described in Sect. S2.6. Observation error bars from Karion et al. (2015) are mostly due to background uncertainty. All enhancements have been calculated first for each downwind segment used in the original MBE analysis of Karion et al. (2015), and then averaged over multiple segments for flights when more than one downwind segment was used in the MBE.**

Similar conclusions apply when looking at the average enhancement in the downwind transects over all eight days (Fig. 4). Despite the spread in model results, most of the Lagrangian model configurations tested consistently showed smaller CH$_4$ enhancements than the observed enhancements, indicating that either the transport and dispersion models are unable to properly capture the influence of emissions on the downwind flight transects, or that the emissions model (the inventory) is too low. The former would

15 be caused by footprints that are too weak, i.e., low sensitivity to emissions. The WRF-Chem model matches the observations relatively better than the others except for 20131019 (due to the high flight altitude relative to the WRF PBL), in contrast to the Lagrangian models, which often underestimate the enhancements regardless of underlying transport. However, on 20131019, WRF-Chem predicts observed enhancements very well if sampled at a lower altitude within the PBL. On 20131028, WRF-Chem



sampled lower in the PBL over-predicts enhancements. The lower altitude WRF-Chem sampling is shown (light pink triangles, Fig. 4) for these two flights only. On 20131020 and 20131025, sampling WRF-Chem at a lower altitude made no difference in the enhancements, as the flight was within the model PBL and it was well-mixed. We note here again that in all of the models shown in Fig. 4, the emissions field is identical (the Z-A inventory); only transport and dispersion are different. The NAM-driven

simulations generally have smaller enhancements than the other models, which can be explained by the generally deeper PBL in that model relative to the other simulations, which results in less sensitivity to surface emissions (i.e., weaker footprints). On 20131025, we suspect that $CH_4$ emissions upwind of the domain likely impacted the downwind measurement, a conclusion also reached by Smith et al. (2015) and Zavala-Araiza et al. (2015), explaining the large underestimate on that day for all models.

### 3.2 Comparison of WRF-Chem and WRF-HYSPLIT forward run $CH_4$ fields

To further investigate differences between the WRF-Chem and Lagrangian models, we use WRF-HYSPLIT forward runs to understand the dispersion of the emitted $CH_4$ in the domain and why the downwind $CH_4$ mole fractions from the two models differ despite identical emissions and meteorological driver data. First, we show very good consistency between the forward-in-time HYSPLIT model simulations and the backward runs (footprint convolutions) (Fig. S6). With good consistency between the forward and backward HYSPLIT dispersion models, the forward WRF-HYSPLIT runs are used to diagnose the differences in

tracer dispersion between WRF-Chem and HYSPLIT. The forward HYSPLIT runs provide a full 4-dimesional $CH_4$ field, unlike the backward runs, which provide simulated $CH_4$ values only along the flight tracks.

We found that $CH_4$ mole fraction enhancements over the domain and throughout the day are generally larger in WRF-Chem, but they are concentrated closer to the ground; in HYSPLIT, the emissions generally mix higher in altitude, especially early in the day,

on most days. Here we focus on two days, 20131019, and 20131028, because they represent the days with very different behavior between the two models.

### 3.2.1    20131019 comparisons

First, we investigate vertical profiles of $CH_4$ mole fraction at a single location downwind of the emissions on 20131019 near the center of the plume as observed by the aircraft, at [32, -97.3]. Figure 5 shows vertical profiles of modeled $CH_4$ (a) and WRF wind

(b) at 16:00 local time, indicating the higher mixing of $CH_4$ in the HYSPLIT model compared to WRF-Chem. The vertical integral of $CH_4$ is slightly smaller in the HYSPLIT model, but the $CH_4$ aloft is being transported out of the domain with faster wind speeds, given the vertical profile of the horizontal wind (Fig. 5b), showing increasing wind speed with height. Thus, the total mass of $CH_4$ emissions advected downwind are the same between the two models, but the combination of higher vertical mixing and faster wind speeds results in a more diluted mole fraction signal in HYSPLIT.






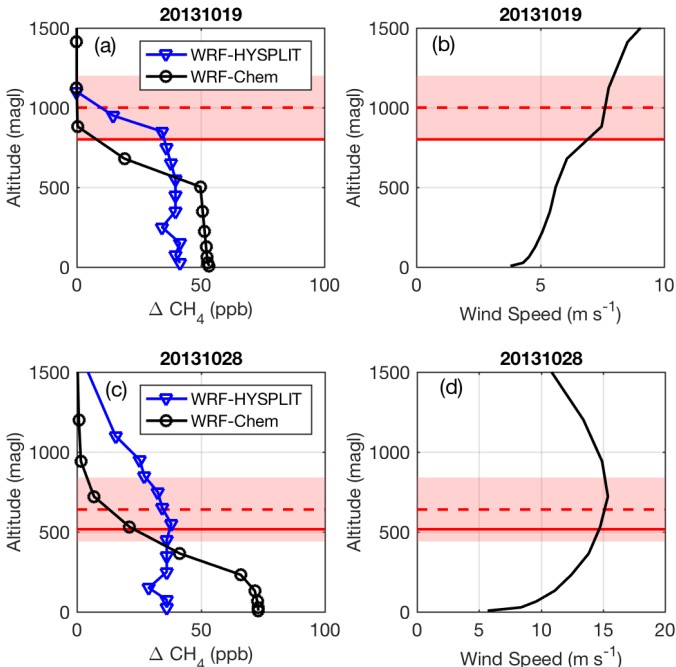

**Figure 5. (a) Modeled CH$_4$ enhancement at a downwind location in WRF-Chem (blue) and WRF-HYSPLIT (black) at 16:00 local time on 20131019 and (c) on 20131028. (b) WRF wind speed profile, for 20131019 and (d) 20131028. In all panels, the WRF PBL depth is the solid red line, the observed PBL is the dashed red line, with the pink shading indicating the uncertainty on the observed PBL (from Karion et al., 2015). Altitudes are given as m above ground level (magl).**

On this day, WRF-Chem shows low enhancements along the flight path (Fig. 2), which was conducted at altitudes between 400 and 900 m above ground level (magl). These altitudes are near the top of the WRF-Chem mixing height, in a zone with a gradient in CH4 (Fig. 5a). However, WRF-Chem enhancements are significantly larger 200 m closer to the ground, due to a shallower WRF mixing depth during flight than observed by the aircraft (31% lower during the flight, Fig. 5a and Table S2). Several vertical profiles indicated that the PBL was growing throughout the flight, averaging about 1000 m during the downwind sampling. In WRF, the PBL is only 800 m at its deepest, and is often even shallower. Indeed, both WRF-Chem and HYSPLIT are unable to replicate the one flight transect at 890 magl (14–14:30 LT in Fig. 2) because it is above the mixed height in both models.

There appear to be competing errors in WRF-Chem for this day. WRF-Chem is trapping emissions closer to the ground than observed, while HYSPLIT mixes emission higher up, closer to the observed PBL height (Fig. 5a). However, wind speed in WRF is also biased high on this day (22% during the flight, Table S1), so the underestimate by WRF-HYSPLIT is likely a combination of factors, including emissions that may be too low and wind speeds that are too high. It is possible that competing errors (low PBL and high wind speed) cancel to allow WRF-Chem to replicate the observations at lower altitudes within its PBL but not above,



while HYSPLIT uses the higher model winds but a higher mixing depth (above the WRF PBL) to dilute the signal. In any case, while HYSPLIT is driven with archived WRF-Chem meteorological fields in this example, the tracer dispersion is quite different from that of WRF-Chem, leading to very different results regardless of which is more accurate.

### 3.2.2    20131028 comparisons

5 The two models perform even more differently on 20131028. On this day, both the WRF modeled winds and observations from High-Resolution Doppler Lidar (HRDL) indicate that winds were slow ($< 2$ m s$^{-1}$) in the morning and sped up throughout the day (Fig. S1). The PBL was also very shallow (~100 m) in the morning hours (Fig. S2). Model-extracted vertical profiles (Fig. 5c) at [34.2, -97.7] at 16:00 local time (21 UTC), a location and time that was sampled by the aircraft downwind of the emissions region, show that the emissions in WRF-Chem are trapped with very high concentrations close to the ground (even more so than on 20131019). WRF also shows a large vertical gradient in wind velocity (Fig. 5d), especially in the lowest 500 m, with stronger winds aloft. CH$_4$ mole fraction profiles throughout the day at this location (Fig. 6, right) show the large difference in vertical dispersion between the two models, with CH$_4$ in WRF-HYSPLIT mixing well above the early morning model PBL, while emissions are trapped in a very shallow stagnant layer near the ground in WRF-Chem.

15 In both models, we investigate the advection of the tracer mass by calculating the hourly change in CH$_4$ mass in a domain surrounding the 25-county area. On this day, WRF-Chem shows a greater mass emission out of the domain during the flight time, while HYSPLIT has a greater mass emission earlier in the day (4–9 local time) and lower during the flight (13–18 local time) (Fig. 6, left). This suggests that the CH$_4$ emissions from the early morning built up and were advected out of the domain later in the day in WRF-Chem, while in WRF-HYSPLIT more CH$_4$ was advected out of the domain by upper-altitude (above the shallow morning 20 PBL) winds in the morning hours. WRF-Chem simulates the vertical dispersion more accurately during the early morning hours in a shallow stagnant boundary layer, as evidenced by its better agreement with flight observations later in the day. We note that in both models the flux out of the emissions region is significantly greater than the true emission rate, i.e., the inventory emission rate of 85 t h$^{-1}$, because of unsteadiness in transport. These differences in vertical mixing are discussed further in Sect. 4.





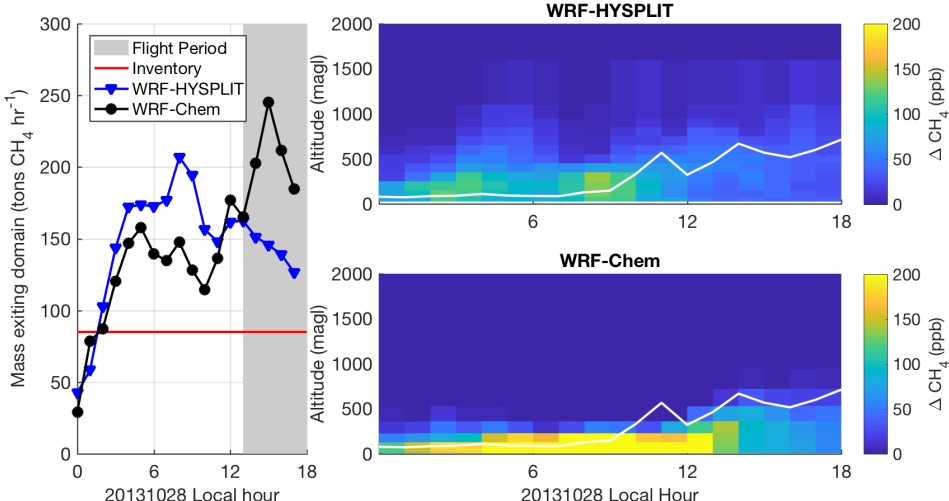

**Figure 6.** Left: CH₄ flux out of the model domain from WRF-HYSPLIT (blue) and WRF-Chem (black) forward runs on 20131028 as a function of time. Red line indicates the constant emission rate from the surface; gray shading indicates flight period. Right: CH₄ enhancement at (34.2, -97.7) as function of height above ground and time of day on 20131028 from WRF-HYSPLIT (top) and WRF-Chem (bottom). White line indicates the PBL depth from WRF-Chem.

### 3.3 Inventory scaling emissions estimates

Given the vertical mixing differences between tracer dispersion models shown above, it is also important to investigate the impact of these differences on estimated CH₄ surface emissions when the models are used to adjust inventory emissions to match the observations. The scaling factor on Z-A inventory emissions (Eq. (1)) for seven transport and dispersion models shows that the CH₄ emission estimates are generally larger than the inventory or MBE results, for most models on many days, while differing significantly between different models and over different days (Fig. 7). We note that these estimates (model results, the MBE, and the inventory) represent total CH₄ emissions, not partitioned among different source categories (i.e., they include natural gas and oil operations as well as emissions from livestock and landfills, etc.). As one would expect from the model predicted CH₄ enhancements of the various Lagrangian models (Fig. 4), the WRF-LPDM dispersion model requires the lowest emissions to explain the observations, while NAM-HYSPLIT requires the highest.

We note here that the MBE result is shown (as a ratio to the Z-A inventory total) for comparison, but is an underestimate of the total 25-county emissions, because many flights did not cover the entire area (more details can be found in Zavala-Araiza (2015).





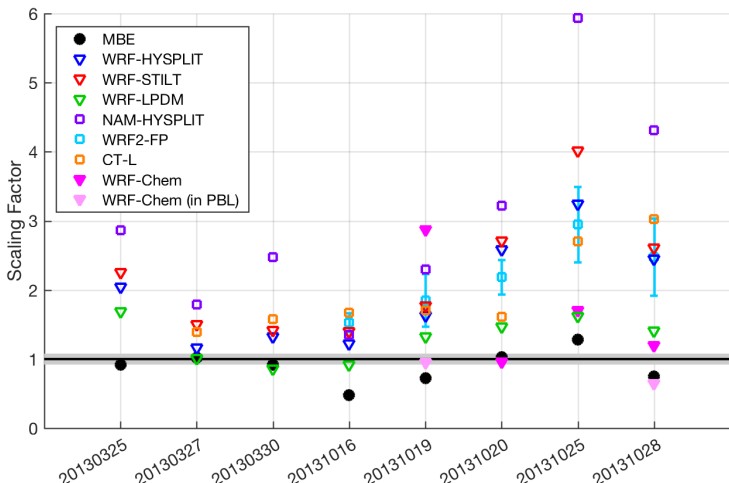

**Figure 7.** Scaling factor on inventory emissions for the various transport/dispersion models (open squares), different dispersion models using identical WRF transport (open triangles), WRF-Chem forward runs (magenta and light pink triangles), and MBE (black circles), compared with the inventory (black line, with gray shading indicating ± 1-sigma uncertainty), for each of eight flights. Error bars on the WRF2-FP runs indicate the standard deviation of results from four different WRF configurations (three for 20131016), as described in Sect. S2.4.

### 3.4 Bayesian inversion results and sensitivity to assumptions

We conducted Bayesian inversions using the flight observations and the WRF-HYSPLIT model, using different choices for the error covariance matrices (Sect. 2.5.2 and S3). Although the sensitivity of the Bayesian posterior to the choice of error covariance matrix construction was considered (Fig. S7), here we discuss the result using RML to estimate a multiplicative factor on the inventory uncertainty for **B**. We assume a single factor on all the grid-cell uncertainty values in the inventory, so that **B** still maintains different values along the diagonal that are proportional to each grid cell's uncertainty. However, we imposed a maximum factor of 10 for the multiplicative factor in the RML, to maintain some realistic confidence in the prior, essentially balancing the effect of the prior vs. the data. The RML optimization tended towards a choice of large **B** variances, likely because the RML method tends to put bias error into **B** rather than **R**, which is assumed to be composed of independent and identically distributed error (Michalak et al., 2005). The maximum value of 10 times the inventory 1-sigma uncertainty was chosen by the RML for all the flights.




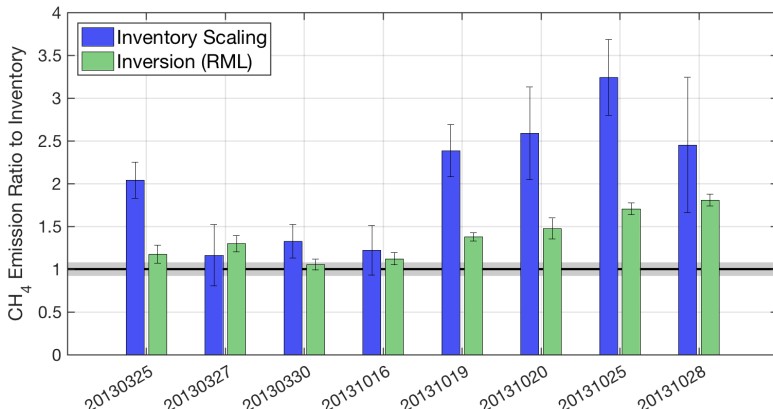

**Figure 8. Summary of total emissions (as a ratio to the Z-A inventory) by flight date (YYYYMMDD), from an atmospheric inversion using RML-derived error covariance matrices, as described in the text, using WRF-HYSPLIT footprints (green). The inventory scaling factor for the same WRF-HYSPLIT model is shown (blue) as a reference. All error bars are k=1 uncertainties. Gray shading around the unit ratio represents the k=1 uncertainty on the inventory.**

Figure 8 shows the results of Bayesian inversions compared with the simple scaling results using the same transport and dispersion (WRF-HYSPLIT), as a ratio to the Z-A inventory, as described in Sect. 2.5.2 and Sect. S3. This comparison of the two flux estimation methods clearly shows that the inversions almost always give lower total posterior emissions than the scaling method. This is reflective of the prior error covariance used for the inversions, which was limited to 10 times the inventory uncertainty (shown in Fig. S8, lower right), but also reflects the fact that the inversions allow for error in the modeled observations (model-data mismatch). The simple scaling method does not incorporate error in the modeled observations into the estimate – it does not have the ability to de-weight observations based on their ability to be simulated accurately by the transport model. In this case study, where transport model error is significant, using an inversion to estimate fluxes is advantageous because it allows the posterior to remain closer to the prior.

In Fig. 8, error bars on the inventory scaling method are from the propagation of error in the choice of background mole fraction from Karion et al. (2015). Error bars on the inversion posteriors are calculated using Eq. (3) summed over the domain and converted to standard deviations. The true uncertainty on the inversion estimate may be significantly larger, as has been shown in other investigations that use inversion ensembles to estimate uncertainty (e.g., Lauvaux et al., 2016), or implied by studies that use an ensemble of forward runs (Angevine et al. 2014), but is not a focus of this work.



## 4 Discussion

The results of the forward model comparison using identical emissions fields indicate differences in the dispersion of the $CH_4$ tracer between the models. Comparison of vertical profiles of the tracer in forward WRF-HYSPLIT and WRF-Chem models shows that the amount vertical mixing of the tracer is very different between these two (Figs. 5 and 6). Given that all the Lagrangian

models (HYSPLIT, STILT, FLEXPART, and, to a lesser degree, LPDM) under-predict enhancements when compared to WRF-Chem (when WRF-Chem is sampled at lower altitudes in the PBL, in the case of 20131019), we expect that all of them parametrize vertical mixing differently from each other and differently from WRF-Chem. Similar discrepancies (although not as large or persistent) between Lagrangian (WRF-STILT) and Eulerian (WRF-Chem) dispersion models were found by Pillai et al. (2012). Ideally the tracer dispersion should follow the underlying meteorological model as faithfully as possible in order to maintain

consistency. For example, it may not be appropriate to mix a tracer in an off-line model above the underlying meteorological reanalysis model (i.e., WRF) PBL height, as the wind speed is often significantly higher above the PBL (and possibly in a different direction due to wind shear).

Flux estimation studies often restrict analysis to mid-afternoon observations (from towers, for example) (e.g., Lauvaux et al., 2016;

McKain et al., 2015) in order to avoid likely model errors during stable overnight conditions. However, our analysis of the flight on 20131028 indicates that in circumstances where the air mass travels over a heterogeneous region with large emissions during conditions that are poorly modeled, there is a substantial effect on the model's ability to simulate those afternoon hours. For example, advection out of the domain in the early morning (4–9 am local time) in the HYSPLIT simulation led to significantly under-predicted concentrations in the late afternoon. This one day may be an extreme example, but without more analysis it is not

apparent how often these kinds of errors may occur and impact flux estimation methods that rely on Lagrangian models to simulate transport and dispersion accurately. If the dispersion model tends to overmix emissions vertically during overnight stable conditions, the additional advection from higher winds aloft could result in a persistent bias in such studies. This result is likely not limited to airborne studies, but the effect may be mitigated in studies using either longer transport time scales (i.e., continental scale studies with footprint time scales of multiple days, because the impact of emissions over longer distances/times is averaged)

or shorter transport time scales of only a few hours, in which observations are not influenced by emissions during times that are difficult to model.

In the current case study, the WRF-Chem vertical parametrization often performed well, but we do not necessarily expect that to be the case across all conditions. Indeed, in one case (20131019), WRF-Chem was unable to predict enhancements at the sampling

altitude of the aircraft due to its low mixing height. However, it is desirable to have the ability to perform an ensemble of dispersion model runs that includes a sufficient range of variability in vertical mixing so that the user can either choose the parametrization that performs best or use the variability in the ensemble to assess meteorological error. Off-line Lagrangian particle models are



typically less computationally expensive and simpler to run than the Eulerian WRF-Chem model. For this reason, future work on the HYSPLIT dispersion model will focus on incorporating additional vertical turbulence parametrizations that are better able to mimic the mixing model in WRF-Chem. In one mixing parametrization that is currently being explored for HYSPLIT, the vertical eddy diffusivity for scalars, $Kz$, is exported directly from the underlying WRF transport and used in HYSPLIT, in an attempt to

5   mimic the mixing in WRF. A preliminary test run with this configuration for 20131028 is shown in Fig. S9 and, although further evaluation of the specific parametrization needs to be performed, the results are promising. This work will be the focus of a future publication that will include testing for this and other case studies.

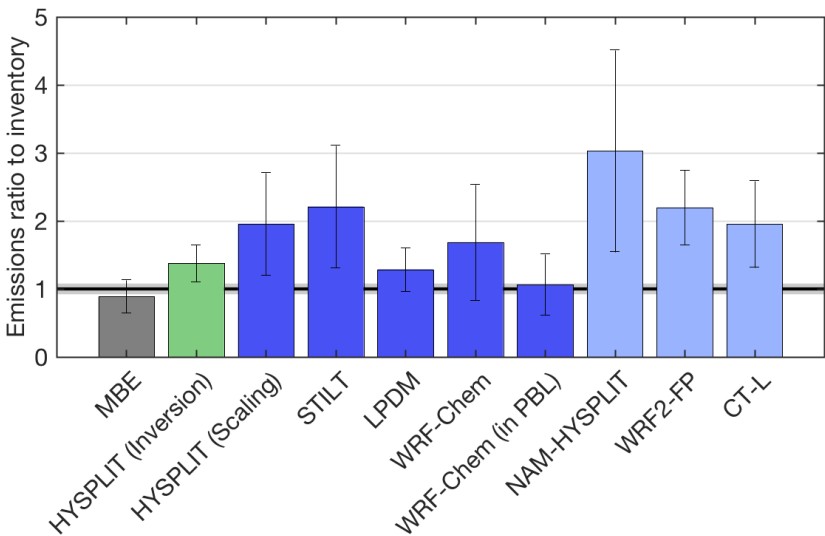

**Figure 9. Mean (± standard deviation in error bars) emissions over all flights available for each transport/dispersion model. MBE (gray) is the result from Karion et al. (2015); HYSPLIT (Inversion)(green) indicates the inversion result; other estimates all use inventory scaling. Dark blue bars indicate dispersion driven by identical WRF meteorology; light blue bars indicate estimates from different meteorology and dispersion. WRF2-FP is the mean of the four different WRF2-FP runs. Shading around the unit ratio indicates the**
15   **reported 1-sigma uncertainty of the total inventory value.**

Using the different transport and dispersion models to estimate total emissions through simple scaling (i.e., adjusting the original emissions map to match observed enhancements), we find that all the Lagrangian-model based estimates are greater than the original inventory, the mass balance estimate from Karion et al. (2015), and the WRF-Chem estimates when WRF-Chem is sampled at the lower altitude for 20131019 and 20131028 (labeled "WRF-Chem (in PBL)" in Fig. 9). More importantly, perhaps, the

20   scaling factors vary significantly between days, as shown in Fig. 7 and by the standard deviations shown as error bars in Fig. 9. Although the MBE results did differ between days (between 0.48 to 1.28 times the inventory, with a standard deviation between





days of 0.24), many of the Lagrangian-model-based scaling estimates vary even more from day to day, with standard deviations ranging from 0.32 (WRF-LPDM) to 1.49 (NAM-HYSPLIT) times the inventory. The simple scaling flux estimation method does not formally account for transport error – fluxes are scaled exactly to match observed enhancements (Kort et al., 2008; McKain et al., 2015). In contrast, the more sophisticated Bayesian inversion framework accounts for model-data mismatch in the final

estimate, most of which is likely caused by transport error. Thus, the inversion posterior estimates are moderated by this error and the optimized fluxes stay closer to the prior (inventory) emissions in most cases, yielding less day-to-day variability. The estimates from the inversion (using WRF-HYSPLIT transport and dispersion) give an average emission rate 1.38 times the inventory, with a smaller standard deviation of 0.27. The inventory scaling method used with the WRF-Chem model gives consistent results for four flights that were simulated with a mean only 2% over the inventory (when two flights were sampled in the PBL instead of the

aircraft altitude) and a low standard deviation between the four flights (0.36 times the inventory). We note these numbers change to 69% over the inventory with a standard deviation of 0.85 when sampled at the aircraft altitude. These results all highlight the importance for vertically-resolved mole fraction observations during model comparison studies.

Given these results, we should consider the possibility that emissions from this region are higher than the inventory, could be

variable in time, and some of the day to day variability observed in emissions estimates could reflect real conditions. First, the models applied in this study suggest that the emissions from the region may be larger than the inventory value, but the magnitude of the difference is certainly unclear given the wide range of model results shown in Fig. 9. Real day to day variability in emissions is likely as well, as reported in a study of emissions from oil and gas extraction in the Eagle Ford Basin, also in Texas (Lavoie et al., 2017). However, the magnitude of the differences in some of the emissions scaling estimates for this Barnett study is unlikely

under any realistic scenario. The emissions from the Barnett are very large in total (85 t h$^{-1}$ in the inventory) and the day to day differences in the top-down estimates can be as large as 50 t h$^{-1}$ (from the MBE or Bayesian methods) or 100 t h$^{-1}$ (from the flux scaling methods). To put this amount into perspective, the hourly rate of $CH_4$ emission from the highly-publicized Aliso Canyon storage facility blow-out in California (Conley et al., 2016) averaged 53 t h$^{-1}$ in the first six weeks. An emissions change such as those shown for some of the models in Fig. 7 would require a blow-out or accident of a similar scale, an event unlikely to go

unnoticed, even in a large dense field like the Barnett. We also note here that three of the eight flights took place on weekend days, with no correlation between weekend days and lower emissions estimates. Some contribution of real variability is of course likely, but most of this variability is likely caused by the inherent uncertainty of these estimation methods and errors in the transport model that depend on daily conditions. The relative consistency in the WRF-Chem results over four flights, and to a lesser extent the WRF-LPDM results, suggest that large variability in some of the other estimates is more likely to be due to dispersion model

error. The high variability in all the emissions scaling estimates indicates that some models exhibit poor performance during such challenging meteorological conditions.



## 5 Conclusion

In this study, we first compared modeled $CH_4$ enhancements from a variety of transport and dispersion models to a set of airborne observations over eight flight days in 2013, using the same underlying emissions map. We then estimated $CH_4$ emissions from the Barnett Shale by scaling the inventory up to match averaged observations using all the models, and finally conducted a simple

Bayesian inversion using one of the models (WRF-HYSPLIT). We found that dispersion model choice had a large impact on the emissions estimate when using the scaling method, with the WRF-Chem Eulerian model matching the original inventory and previous MBE estimates best, with the caveat that it sometimes trapped emissions closer to the surface than observed during flights. As discussed in Sect. 4, disagreement between results using the same underlying meteorological model indicates that more focus should be put on tracer dispersion modeling, specifically with regard to vertical mixing, in the off-line dispersion models.

Our analysis of this data set indicates systematic differences in vertical mixing among different tracer dispersion models even while using the same underlying meteorology from WRF, with generally more vigorous mixing in the Lagrangian models, although to different extents. We have also found that incorrect vertical mixing of a tracer during overnight hours and stable PBL conditions can have a significant effect on tracer concentrations several hours later in the day. We also note that inconsistencies between the

vertical mixing parametrization of the off-line Lagrangian dispersion model and the meteorological model driving it are a more general cause of errors and the coupling between Eulerian transport and Lagrangian dispersion models should be investigated further. However, more research into these errors would need to be conducted to determine how wide-spread these kinds of biases might be if the analysis was carried out over an entire season or year, for example. It would also be useful to consider the extent to which these findings are relevant for Lagrangian footprint calculations spanning multiple days as have been done for continental-

scale inversions (e.g., Gourdji et al. (2010); Schuh et al. (2013)).

### Data availability

Aircraft observations used in this study are available at NOAA/ESRL (web link TBD). Meteorological (WRF) and tracer dispersion model output is available on the NIST data portal (data.nist.gov [link TBD]). The Z-A $CH_4$ inventory (Zavala-Araiza,

2015) is available by request from the authors of that study. Researchers are encouraged to contact the authors for any additional data or advice on the use of this data set.

### Acknowledgements

The authors would like to thank Tim Bonin, Mike Hardesty, and Alan Brewer at NOAA/ESRL Chemical Sciences Division for providing HRDL data products; Marikate Mountain (AER) for Carbon Tracker-Lagrange (CT-L) WRF-STILT model runs; Eric

Kort (University of Michigan), Paul Shepson (Purdue University), and the teams from NOAA/ESRL, University of Michigan, and



Purdue for collaborating on the Barnett flight campaign and collecting airborne data; Daniel Zavala-Araiza and David Lyon for providing the Barnett CH₄ inventory; and Fantine Ngan for assistance in development of the experimental HYSPLIT parametrization. Certain commercial equipment, instruments, or materials are identified in this paper in order to specify the experimental procedure adequately. Such identification is not intended to imply recommendation or endorsement by the National

5   Institute of Standards and Technology, nor is it intended to imply that the materials or equipment identified are necessarily the best available for the purpose.





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
