# Peer review of "Inter-comparison of Atmospheric Trace Gas Dispersion Models: Barnett Shale Case Study"

_Atmospheric Chemistry and Physics, 2018_

## Referee Comment (RC1) · Anonymous Referee #2 · 24 Nov 2018

General remarks This is an expert study of a difficult and important problem. The paper is well written and the logic is clear. It should be published, and only very minor revision is needed.

Quantifying trace gas fluxes by using atmospheric measurements around known and unknown sources is very difficult. The problem is extremely important, especially for greenhouse gases. Mitigating emissions, especially methane, will depend on prioritizing the sources to be targeted, for example in a gasfield, so that the most cost-effective reduction measures can be carried out. But the problem is that it is very difficult to get accurate flux estimates from airborne (and indeed, also vehicle-borne) atmospheric measurement, so it is hard to quantify and thus rank emissions.

Karion et al. address this important problem by comparing a variety of modeling ap-

proaches, testing them against observational data, and seeking out the weaknesses in the different modeling approaches. The focus is on methane. This is an extremely important problem. There is a substantial gap, both globally and in major emitting regions, between bottom-up inventories of emissions and top-down flux quantification. Curiously, globally the bottom up estimates are much higher; regionally, top-down assessments tend to be greater.

The paper begins with a strong introduction to the problem: most current flux assessments rely on very simple mass balance methodologies, and are likely both imprecise and inaccurate. Karion et al. investigate the use of meteorological and tracer dispersion models, tested against observations. In particular, they find that the models differ substantially in their vertical dispersion, and their work points to the need for better understanding of vertical mixing in calculating regional inversions.

Karion et al. are fortunate in their wealth of observational data and in the power and choice available for their tracer dispersion modeling. Most teams studying this problem worldwide, especially in tropical countries, do not have access to these sophisticated techniques. Aircraft are expensive but UAV measurement by grab-bag or lofted hose is becoming feasible, and simple HySPLIT trajectory analysis becoming accessible even to school-project landfill studies in tropical Africa. With luck, as UAVs evolve and dispersion models become more accessible, work such as this study by Karion et al. will lead eventually to the spread of accurate modeling by less resourced teams in less-developed regions. Eventually, that may close the top-down vs bottom-up gaps.

The study is well defined, the methodology is well explained, the analysis is thorough, and the findings are both convincing and useful. The paper should be published with minor changes.

Trivial points Page 4 Section 2.1 line 1. Figure 1 shows the regional map, or the map of the region, not the "region map". Page 9 line 7 give the year for Karion et al. References: generally, use et al. when there is a telephone book of austhors. For Lauvaux

et al. and also for Pillai et al, use $CO_2$, not format script. For Stein et al, decapitalise title.

Conclusion Accept, after very minor technical revisions
* * *

---

## Referee Comment (RC2) · Anonymous Referee #1 · 27 Nov 2018

Review Comments: Karion et al, 2018.

I enjoyed this paper, and I feel that this manuscript brings together some excellent work highlighting some of the potential pitfalls of emission flux modelling using dispersion models. I recommend publication of this work, subject to some very minor corrections / edits.

The paper is well structured and very readable, with enough information generally being given to satisfy both the observational and modelling communities.

My most significant negative comment regards the current state of the conclusions. I feel that on their own they do not really show off the depth or value of the work presented. Currently, the conclusions hide behind the "we need to more research"

[Figure]

caveat, whereas I feel the manuscript has the detail within the results and discussion to put something more brave and definitive into the conclusions. I would ideally see this as some sort of recommendations / best practice for selection and set up of model given these circumstances. I'm sure that I'm not alone in reading papers "Abstract-Conclusions-Discussion-Other" and a stronger conclusion will draw more readers into the meat of the manuscript.

\*\*\*\*\*\*\*\*\*\*\*\*\*\*\*\*\*\*\*\*\*\*\*\*\*\*\*\*\*\*\*\*\*\*\*\*\*\*\*\*\*\*\*\*\*\*\*\*\*\*\*\*\*\*\*\*\*\*\*\*\*\*\*\*\*\*\*\*\*\*\*\*\*\*\*\*\*\*\*\*\*\*\* Very minor comments:

Abstract: It would be good to have the models used in the investigation named in the abstract. L31: Can you put a more definitive number rather than "significant" to describe the variability.

Methods:

P3. L3. Have any other models been used to successfully simulate CH4 observations? From the way it is written it makes it seem that WRF-chem is the only one thought to be capable.

P4. L17. Why use the Texas Railroad definition? Is there not a more appropriate definition from the USGS or similar?

P6. L9. "The footprints…" Could this be summarised into an equation form? It feels quite a wordy explanation of the calculation process? The same comment also applies to P7. L13-15.

Note, editorial. Some of the equations do not print properly (e.g. P6. L9), despite it being in pdf format. There may be an issue with the way the equations are encoded, please check as this may be an issue local to my computer or a more general issue.

P7. L6. The CH4 inventory was provided at 0.1 degree resolution, this reads as if there were options on the resolution? Were there options, and if so why was this chosen?

P9. L6. Could a small paragraph be added to explain how the errors from the transport model would be expected to influence the data and what magnitude of errors are expected to exist from the transport model?

Results:

P11. Figure 2. I find it very difficult to see the difference in the colours between NAM-HYSPLIT and WRF-HYSPLIT. Could you look into changing the colour scheme to make them all more distinctive (same applies to other figures where same colour scheme used)

Discussion:

P20. L4. Can you please define "very different" with a numerical value

P20. L19-26. This feels like a very important part of the discussion. Is there any recommendation / specific experimental set up that could be recommended which could help resolve this?

P22. L12. This is excellent, and I feel should be given more prominence in the conclusion.

Conclusion:

See comment at start of review.

---

## Author Comment (AC1) · 1 Feb 2019

Author response to reviewers for article submission to ACP:

**"Inter-comparison of Atmospheric Trace Gas Dispersion Models: Barnett Shale Case Study", Karion et al.**

We would like to thank both reviewers for reading and reviewing our manuscript. We have responded to the minor revisions requested below and in the revised text. In the responses below, the reviewer comments are in regular text and our response is italicized below each comment.

**Reviewer 1**

*We thank the reviewer for the positive comments overall. We agree that dispersion modeling is an important topic to highlight at this time as the use of atmospheric transport and dispersion models becomes more widespread for top-down emissions verification.*

Page 4 Section 2.1 line 1. Figure 1 shows the regional map, or the map of the region, not the "region map".

*This text has now been changed to "a regional map".*

Page 9 line 7 give the year for Karion et al.

*This has been fixed, in this location and elsewhere where the year was missing.*

References: generally, use et al. when there is a telephone book of authors.

*We do have many references with long author lists. We have followed the reference formatting found online in the ACP guidelines, which requires the full list of authors. We will consult with the editorial staff on the proper format if it needs to change.*

For Lauvaux et al. and also for Pillai et al, use $CO_2$, not format script. For Stein et al, decapitalise title.

*Both of these have been corrected in the revision.*

**Reviewer 2**

*We thank the reviewer for the positive review and thoughtful suggestions and comments.*

My most significant negative comment regards the current state of the conclusions. I feel that on their own they do not really show off the depth or value of the work presented. Currently, the conclusions hide behind the "we need to more research" caveat, whereas I feel the manuscript has the detail within the results and discussion to put something more brave and definitive into the conclusions. I would ideally see this as some sort of recommendations / best practice for selection and set up of model given these circumstances. I'm sure that I'm not alone in reading papers "Abstract- Conclusions-Discussion-Other" and a stronger conclusion will draw more readers into the meat of the manuscript.

*This is a good point and we have now strengthened the conclusions by indicating future work on this problem that is planned, with specifics on how we are going to be testing this same set of dispersion models against known releases. We have added an additional final paragraph to emphasize our key take-away point:*

*"In short, vertical mixing in the dispersion models differs despite being driven by identical meteorological fields, leading to different footprint strengths between the models and thus different flux estimates. Thus, evaluating only the meteorological variables such as wind speed, wind direction, and PBL depth is not sufficient for assessing transport error. Using a single model that might be biased under certain conditions can especially bias flux estimates when using a limited number of independent observations (e.g. eight flight days). We strongly recommend that multiple transport and dispersion models be used when conducting this type of analysis of atmospheric observations."*

Abstract: It would be good to have the models used in the investigation named in the abstract.

*We appreciate this point, but due to the length of the model name list (the acronyms would need to be spelled out), we have chosen not to include the models in the abstract, as it would impact its readability. We believe that the list in Table 1 is sufficiently accessible to readers that might want to skim the paper and find the model names easily without going through all the text.*

L31: Can you put a more definitive number rather than "significant" to describe the variability.

*We have added a parenthetical now indicating that we found variability up to a factor of three between models.*

Methods:
P3. L3. Have any other models been used to successfully simulate CH4 observations? From the way it is written it makes it seem that WRF-chem is the only one thought to be capable.

*We have modified this sentence to specify that WRF-Chem has been used at similar regional and local spatial scales to simulate CH4. Here we are referring only to Eulerian models, and we are not aware of other Eulerian models that have been used to simulate CH4 observations at these smaller spatial scales (i.e. not global-scale).*

P4. L17. Why use the Texas Railroad definition? Is there not a more appropriate definition from the USGS or similar?

*We remain faithful to the Texas Railroad Commission definitions of the Barnett region from the previous literature on this campaign, i.e. both the inventory papers (Lyon et al., Zavala-Araiza et al.), and the previous definition for the mass balance estimate (Karion et al 2015). The Texas RRC is the agency reporting all information on Texas oil and gas production and wells.*

P6. L9. "The footprints. . ." Could this be summarised into an equation form? It feels quite a wordy explanation of the calculation process? The same comment also applies to P7. L13-15.

*To clarify this, we have now added new Equations 1 and 2 at the first description and refer to them again on Page 7, and also add another reference to Lin et al 2003 where this calculation is described in significantly more detail.*

Note, editorial. Some of the equations do not print properly (e.g. P6. L9), despite it being in pdf format. There may be an issue with the way the equations are encoded, please check as this may be an issue local to my computer or a more general issue.

*Thank you for this note, we will consult with the editorial staff to make sure these are posted properly in the final version.*

P7. L6. The CH4 inventory was provided at 0.1 degree resolution, this reads as if there were options on the resolution? Were there options, and if so why was this chosen?

*The original Z-A inventory was available to us at 4-km and 0.1-degree resolution. We used the 0.1 degree resolution to save computation time in calculating the footprints and estimating fluxes using the inverse model. We tested the 4-km inventory as well by gridding to 0.04 degrees for the worst-performing flight (10/28) to test whether higher resolution would improve the results but found no impact at all in the simulation of CH4. (This is now mentioned in table 1 and SI). We have modified the text here as well to mention this.*

P9. L6. Could a small paragraph be added to explain how the errors from the transport model would be expected to influence the data and what magnitude of errors are expected to exist from the transport model?

*We have lengthened this sentence to explain the direct relationship between error in modeled CH4 and the scaling factor. These errors are not often known or identifiable ahead of time, which is why people have sometimes used ensembles of transport models to assess their differences as a proxy of uncertainty. Here we evaluate the WRF wind speed and PBL depth in the SI, and investigate the differences between this ensemble of meteorological simulations and dispersion models. We believe further discussion of how to incorporate transport error into a simple scaling analysis is outside the scope of this work.*

Results:

P11. Figure 2. I find it very difficult to see the difference in the colours between NAM-HYSPLIT and WRF-HYSPLIT. Could you look into changing the colour scheme to make them all more distinctive (same applies to other figures where same colour scheme used)

*We have now changed the color of the NAM-HYSPIT model run results in Figures 2, 3, 4, and 7 to be a lighter purple, so it will not be confused with the dark blue for WRF-HYSPLIT.*

Discussion:

P20. L4. Can you please define "very different" with a numerical value

*This has been added to the text.*

P20. L19-26. This feels like a very important part of the discussion. Is there any recommendation / specific experimental set up that could be recommended which could help resolve this?

*This is a good point, but I am not entirely sure how we could look at the possible bias caused in a long-range or regional simulation from incorrectly modeling stable conditions during night-time. Tracer experiment data over long transport times would need to be used, but it would be difficult if using a powerplant plume for example as a known emitter because after a days' transport the emissions will have diluted and mixed to make them difficult to measure. The short time scale would be an easier test of course. Historical tracer data is available that covers continental scales (NOAA/ARL's CAPTEX experiment, for example), using tracers that are not present in the atmosphere already. We now have added a sentence to the text suggesting that tracer experiments could be used at the proper length and time scales to investigate this issue of nighttime simulations.*

P22. L12. This is excellent, and I feel should be given more prominence in the conclusion.

*We agree that this was not highlighted adequately, and a sentence is now added in the conclusion pointing out the need for vertically resolved measurements of mole fraction, wind, and turbulence.*